# Indoor Localization Using Positional Tracking Feature of Stereo Camera on Quadcopter

**Ahmad Riyad Firdaus, Andreas Hutagalung, Agus Syahputra and Riska Analia ***

Department of Electrical Engineering, Politeknik Negeri Batam, Kota Batam 29461, Indonesia
* Correspondence: riskaanalia@polibatam.ac.id

**Abstract:** During the maneuvering of most unmanned aerial vehicles (UAVs), the GPS is one of the sensors used for navigation. However, this kind of sensor cannot handle indoor navigation applications well. Using a camera might be the answer to performing indoor navigation using its coordinate system. In this study, we considered indoor navigation applications using the ZED2 stereo camera for the quadcopter. To use the ZED 2 camera as a navigation sensor, we first transformed its coordinates into the North, East, down (NED) system to enable the drone to understand its position and maintain stability in a particular position. The experiment was performed using a real-time application to confirm the feasibility of this approach for indoor localization. In the real-time application, we commanded the quadcopter to follow triangular and rectangular paths. The results indicated that the quadcopter was able to follow the paths and maintain its stability in specific coordinate positions.

**Keywords:** quadcopter; ZED 2 stereo camera; indoor localization; real-time application

## 1. Introduction

Research into unmanned aerial vehicles (UAVs), for example, the quadcopter type, has various purposes, including military applications [1], wildlife monitoring [2], agriculture [3], civilian purposes, for example, for the delivery of payload [4], or the recording of unreachable scenery [5]. With respect to configuration, a quadcopter can be categorized as "+" and "x" [6]. Even if the configuration is different, the quadcopter's performance is the same. To control this kind of UAV, researchers have introduced different types of control methods, including Fuzzy+PID, to control the attitude of an octocopter with the same configuration as a quadcopter [7], sliding mode control-based interval type-2 fuzzy logic [8], linear quadratic regulators (LQR) [9], fuzzy logic [10], H∞ control [11], adaptive control [12], gain scheduling [13], backstepping control [14], and PID control [15].

When a quadcopter has been controlled for the attitude and altitude, it can be ordered to maneuver by following a path or self-localizing in indoor or outdoor applications. For example, [16] used a GPS sensor to decentralize the localization to detect multiple quadcopters. Another localization method is the SLAM, which utilizes a Lidar sensor to reconstruct a 3D scene [17], or uses a camera to extract feature-based direct tracking and mapping (FDTAM) information to reconstruct a 3D scene in a real-time application [18]. Moreover, for indoor localization, [19] introduced the method called the time of arrival (ToA) or the time difference of arrival (TDoA), eliminating noise using a Gauss—Newton approach. This method resulted in a good precision. However, it generated acoustic noise. Another method, described in [20], introduced visual odometry by utilizing stereo vision; however, the visual odometry employed was sensitive to lights.

In [21], a hybrid acoustic was used based on time-code division multiple access (T-CDMA) and an optical module of a time-of-flight (TOF) camera for indoor positioning; however, the method uses an acoustic module to perform a 3D multilateration to estimate the position of the drone, which involves a significant calculation. To reduce the calculation

involved for estimating the object coordinate, a ZED 2 stereo camera [22] can be used as this camera is equipped with positional tracking coordinates. Based on this feature provided by the stereo camera, in this investigation, we used a stereo camera to perform indoor localization. We evaluated the method used through experiments using real-time applications with different path patterns.

## 2. Related Work

The development of UAV navigation systems has been investigated for both outdoor and indoor applications. For outdoor applications, the global positioning system (GPS) sensor can be relied on to understand the environment surrounding the UAV during navigation. In [23], a GPS sensor was used to track waypoints based on a robot operating system (ROS) along with autopilot sensors, and a dense optical flow algorithm which were integrated for hovering and tracking in an outdoor environment. In [24], inertial devices and a satellite navigation system were combined to improve the fusion positioning, accuracy, and robustness. Another investigation used the Global Navigation Satellite System (GNSS) for an outdoor navigation system [25,26]. With respect to indoor applications, the GPS sensors are not able to transmit their position signal through a building; therefore, many investigators have introduced indoor localization methods utilizing several sensors. In [27], the fusion of a Marvelmind ultrasonic sensor and a PX4Flow flow camera was used to measure the position and optical flow for indoor navigation based on a robotic operation system (ROS).

Furthermore, in [28], an optical flow and Kalman filter were used to estimate the camera's position, then semantic segmentation was performed based on deep learning to determine the wall position in front of the drone. The authors of [29] employed the RFID received signal strength and sonar value to perform a localization in indoor applications, in cooperation with a vision system for landing procedures. In [30], a LIDAR-based 2D SLAM was used to enable the drone to understand the environment surrounding it in a simulation using MATLAB. Other investigators have sought to combine a stereo camera and ultrasonic sensor to detect a surrounded object and extract three-dimensional (3D) clouds for path planning. However, this method involves a limited field of view when handling the UAV movement; the system needs to change the heading before moving side-to-side or backwards, and, in addition, potentially requires heavy computational resources [31].

In contrast to [31], the present investigation was performed using only a single ZED 2 stereo camera to perform an indoor localization or navigation based on the features given by the camera. To use the stereo camera for an indoor localization, we first transformed the coordinates from the camera into a NED coordinate system. Then, we used this coordinate to estimate the position and maintain the stability of the UAV during hovering.

## 3. Materials and Methods

This section will describe the materials and methods used in this work in two parts: the coordinate transformation and the system architecture. The coordinate transformation will explain how to convert the coordinate from a rigid body perspective to the camera coordinate system. Then, the system architecture will explain the architecture of the drone to achieve an indoor localization, which consists of the stabilization control mode and the pose estimation system.

### 3.1. The Coordinate Transformation

In this work, we utilized the ZED 2 camera to help our quadcopter to maintain its position in a particular coordinate. One of the features of the ZED 2 camera is positional tracking, consisting of an IMU sensor. The positional tracking coordinate provided by the ZED 2 camera has the opposite direction as the quadcopter. We have to transform the camera coordinate into the NED system of the quadcopter. To convert this coordinate, first we need to derive it from the rigid body of the quadcopter. The rigid body rotation illustrated in Figure 1 represents the distance between two points, the indicate point and

the total point the quadcopter has, as presented in Equation (1); to obtain the displacement orientation point, we set this output of this equation with a constant number. Additionally, we can assume the position in the 3D coordinate (X, Y, Z) as presented in Figure 2.

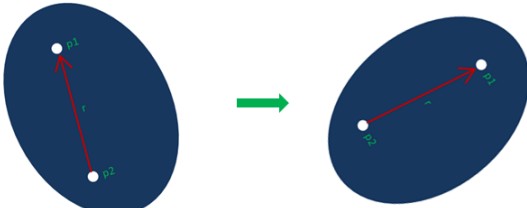

**Figure 1.** The rigid body position of r when it rotated to a certain angle.

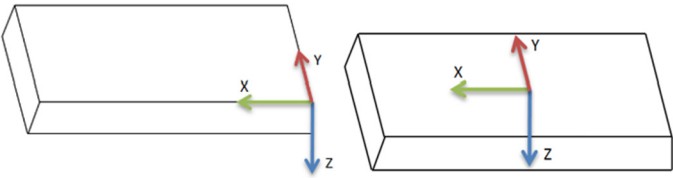

**Figure 2.** The illustration of the 3D coordinate system in the rigid body.

In order to determine the position in 3D coordinates, we set the frame of the reference as {O} and the body frame was represented as {B}. Because the body frame of the drone will change during the flight, the position can be obtained by comparing the body frame to the reference frame, as illustrated in Figure 3. In developing the quadcopter, we need to understand its orientation due to the rotation movement of the drone. We can describe the direction of a 3D rigid body by using the rotation matrix and Euler angle. The rotation matrix of the drone derives from Equations (2)–(4), and if we transpose the matrix using the left-hand rule, the equation becomes Equation (5).

$$\left| r_{p[i]} \right| = r_{p[i]} = constant \tag{1}$$

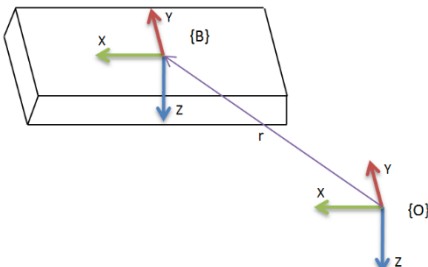

**Figure 3.** The coordinate position of the body frame and reference frame during displacement.

Another, in developing the quadcopter, we need to understand its orientation due to the rotation movement of the drone. We can describe a 3D rigid body's orientation using the rotation matrix and Euler angle. The rotation matrix of the drone is derived from Equations (2)–(4), and if we transpose the matrix using the left-hand rule, the equation is represented as Equation (5).

$$R_x(\theta) = \begin{bmatrix} 1 & 0 & 0 \\ 0 & \cos\theta & -\sin\theta \\ 0 & \sin\theta & \cos\theta \end{bmatrix} \tag{2}$$

$$R_y(\theta) = \begin{bmatrix} \cos\theta & 0 & \sin\theta \\ 0 & 1 & 0 \\ -\sin\theta & 0 & \cos\theta \end{bmatrix} \tag{3}$$

$$R_z(\theta) = \begin{bmatrix} \cos\theta & -\sin a\theta & 0 \\ \sin\theta & \cos\theta & 0 \\ 0 & 0 & 1 \end{bmatrix} \tag{4}$$

$$R_x(\theta)^T = \begin{bmatrix} 1 & 0 & 0 \\ 0 & \cos\theta & \sin\theta \\ 0 & -\sin\theta & \cos\theta \end{bmatrix} \tag{5}$$

From Equations (2)–(4), we can represent the orientation using the Euler angle rotation matrix. The Z, Y, and X transformed into $\alpha$, $\beta$, and $\gamma$, derived from the ZYX Euler angle in Equation (6); therefore, the rotation matrix in each axis is represented in Equations (8)–(10), where Equation (7) denotes the identity matrix.

$$R(\alpha, \beta, \gamma) = I \, \mathrm{Rot}\left(\vec{z}, \alpha\right) \cdot \mathrm{Rot}\left(\vec{y}, \beta\right) \cdot \mathrm{Rot}\left(\vec{x}, \gamma\right) \tag{6}$$

$$I = \begin{bmatrix} 1 & 0 & 0 \\ 0 & 1 & 0 \\ 0 & 0 & 1 \end{bmatrix} \tag{7}$$

$$\mathrm{Rot}\left(\vec{z}, \alpha\right) = \begin{bmatrix} \cos a & -\sin a & 0 \\ \sin a & \cos a & 0 \\ 0 & 0 & 1 \end{bmatrix} \tag{8}$$

$$\mathrm{Rot}\left(\vec{y}, \beta\right) = \begin{bmatrix} \cos\beta & 0 & \sin\beta \\ 0 & 1 & 0 \\ -\sin\beta & 0 & \cos\beta \end{bmatrix} \tag{9}$$

$$\mathrm{Rot}\left(\vec{x}, \gamma\right) = \begin{bmatrix} 1 & 0 & 0 \\ 0 & \cos\gamma & -\sin\gamma \\ 0 & \sin\gamma & \cos\gamma \end{bmatrix} \tag{10}$$

This section aims to obtain the coordinate transformation from the camera view to the NED system, illustrated in Figure 4. In this work, we transform the coordinate using the homogeneous transformation, represented in Equations (11)–(13).

$$H = \begin{bmatrix} R & d \\ 0 & 1 \end{bmatrix} \tag{11}$$

$$R = \begin{bmatrix} i_x & j_y & k_z \\ i_x & j_y & k_z \\ i_x & j_z & k_z \end{bmatrix} \tag{12}$$

$$d = \begin{bmatrix} d_x, & d_y, & d_z \end{bmatrix}^T \tag{13}$$

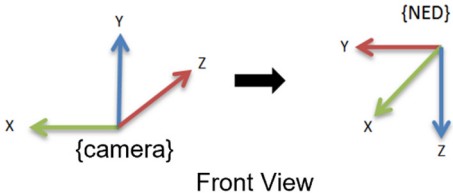

**Figure 4.** The illustration of transforming the camera coordinate to the NED system.

### 3.2. The System Architecture

The system architecture of our proposed method is illustrated in Figure 5. In this work, we utilized the ZED 2 stereo camera, which is equipped with the positional tracking feature. We also employed the Jetson Nano onboard computer to process the coordinate transformation, detect the object, and send the coordinate estimation to the flight controller.

The Pixhawk Cube Blank has been chosen as the flight controller to control the quadcopter orientation. This flight controller provided the ground control system software to witness the data transmission from the quadcopter movement and the camera input.

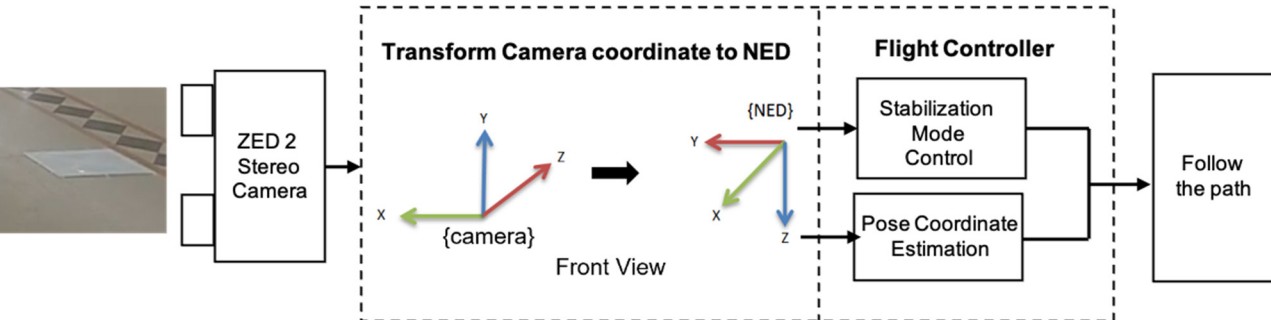

**Figure 5.** The system architecture of indoor localization.

In order to estimate the pose and let the quadcopter achieve a self-localization indoors, the camera will first detect the white square paper on the floor. Then, we collect the position, orientation, and confidence level from this sample image. Then, all this information collected from the camera will be sent to the Jeston Nano to process the coordinate transformation. The orientation and position data from the camera sent consists of 6 degrees of freedom: x, y, and z for the position vector and w, x, y, and z for the orientation. The process of transforming the camera coordinate into the NED frame is illustrated in Figure 6. This process was carried out on the onboard computer, where the number (2) from Figure 6 denoted the pose and confidence level data which is generated by the camera. Numbers (3) and (1) represented the offset configuration of the camera towards the flight controller to estimate the distance. To convert the camera coordinate into the NED frame, we utilized the homogenous transformation matrix, as presented in equation (11), by multiplying the value of offset camera configuration (1) and (3) with the pose and confidence level data.

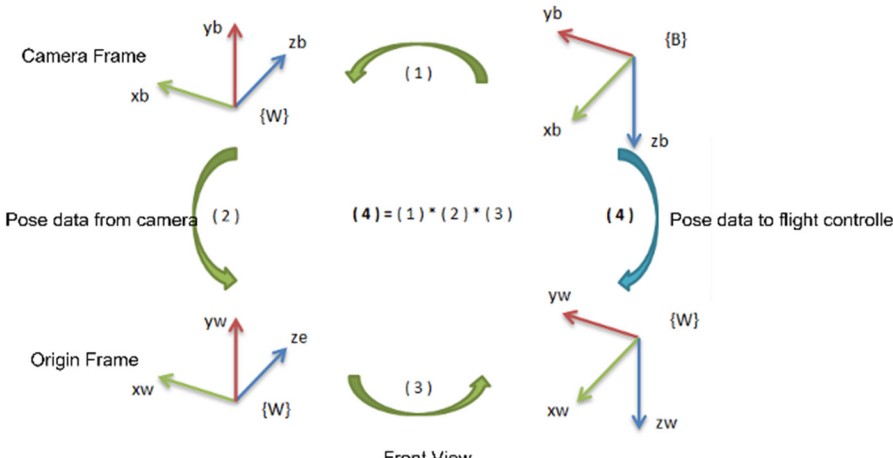

**Figure 6.** The illustration of the coordinate transformation process.

The results of this calculation are the rotation and translation of the quadcopter, which is represented in number (4).

When the coordinate transformation is ready, the data will be sent to the flight controller to activate the stabilization mode control. Because the flight controller is already firm regarding the attitude, we only send the camera's position and orientation. The altitude will be held at a particular position, which depends on the position data. In addition to the mode control, this data has also been used in the pose estimation. To make the drone follow

the white square paper path on the floor, we combined the mode control stabilization and the pose estimation on the flight controller to ensure that the drone could move according to the path given.

## 4. Results

The experiment in this work was carried out in a real-time application. At first, we did experiments to understand how stable our system was when we pushed the drone using a rope. Then, we performed the indoor localization to follow the square and triangular paths. All the coordinates given in this experiment were measured in meters and were plotted in the cartesian mode.

### 4.1. The Stability Experiments

In the scheme for verifying the stability performance, we commanded the drone to remain hovering in some coordinates and to land at the same place after the remote control was released. In this experiment, the drone should stay at coordinate $(0, -0.5)$—this experiment's illustration is presented in Figure 7. As shown in Figure 7, first, we flew the drone using a remote control and commanded the drone to stay hovering at a specific position. After we released the remote control, the drone moved to the left and landed at the end of the track. This movement indicates that the drone cannot hover in the same place for some time.

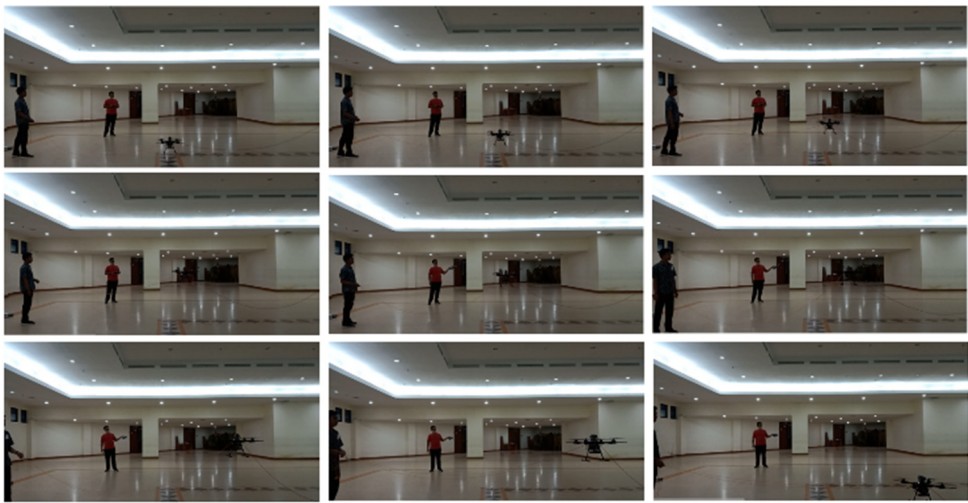

**Figure 7.** The drone movement without activating the stability mode control.

The result of this experiment is represented in Figure 8. As seen in Figure 8, the graph was divided into three parts. The first graph was the drone position while hovering, printed in cartesian view, where the green line represented the track of the drone while hovering, the blue dot represented the take-off position, and the red dot denoted the landing position. The second one was the average error of X and Y generated by the drone movement, the red line represented the error on the X dan Y position, and the orange one indicated the average error of both positions. Additionally, the last was the time response of the X dan Y during the flight in a certain position, which has a blue line for the time response of the X and Y movement, and the orange represents the setpoint.

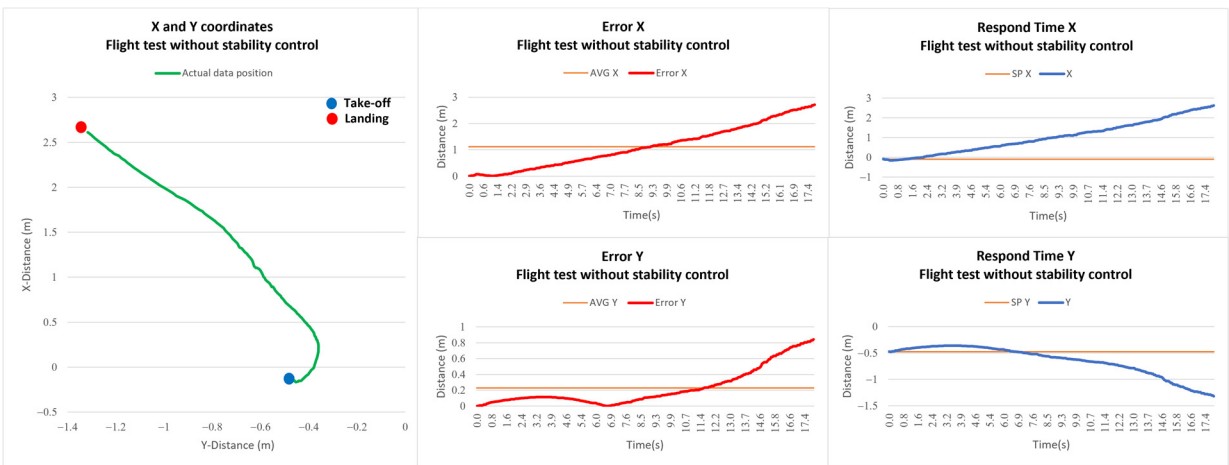

**Figure 8.** The response results from drone movement without activating the stability mode control.

From Figure 8, we can see that the drone was first taken off at the coordinate 0 m on the X-axis and about −0.5 m on the Y-axis. However, it dodged from the original position to the coordinates of 2.5 m on the X-axis and −1.4 m on the Y-axis. Additionally, then, the drone landed at these coordinates. From this movement, the average error produced is about 1 m on the X-axis and 0.2 m on the Y-axis. For the response system, the drone can only stay at coordinate 0 on the X-axis for about 2.4 s and the Y-axis for around 7 s. In this experiment, the drone failed to maintain its position if we turn off the control position mode.

Another stability experiment was conducted when the stability mode control was activated. In this experiment, we did the same scheme as presented in Figure 7, only in this experiment the coordinate position is about (0.08, −0,2) and lets the drone hover in this position. The result of this experiment is illustrated in Figure 9, where the drone was able to maintain its position, depicted on the first graph in Figure 9. The average error generated by the drone's movement is about 0.05 m for both the X- and Y-axes. The results of the time response position on the X and Y coordinate can be seen on the last graph, where the response results show that the drone was able to maintain its position with an acceptable error.

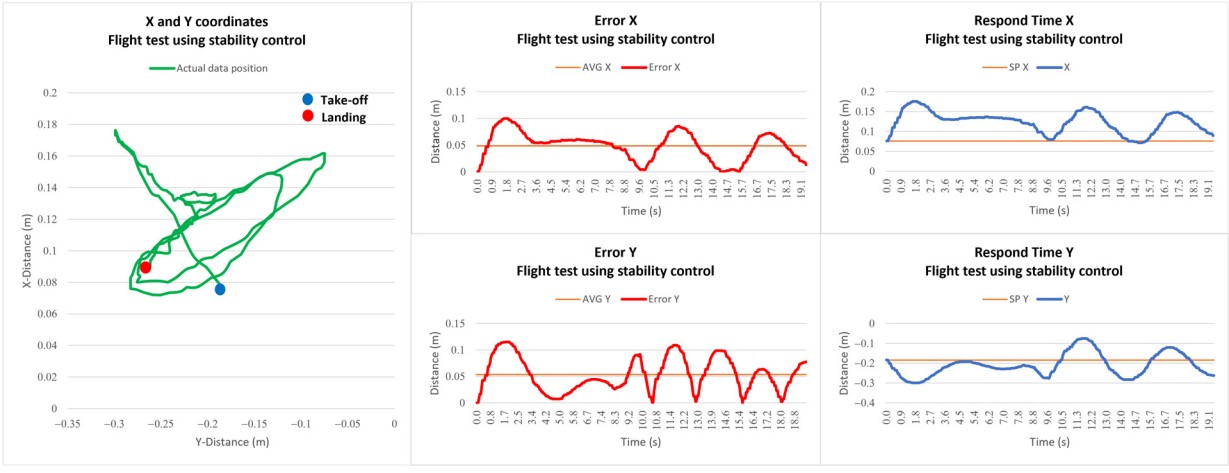

**Figure 9.** The response results from drone movement when activating the stability mode control.

To ensure our proposed position control worked well, we also compared the response results when we pulled the drone with the rope, as illustrated in Figure 10. In this experiment, we first flew the drone at some height in a particular coordinate. After the drone hovered, we pulled the rope to the right, as presented with the blue arrow, and the drone

followed the rope path afterward; however, it landed away from its original position to the left. As for the response result represented in Figure 11, the original coordinate before the drone was pulled out is (0,0), yet after pulling out, the drone moved away from its original position and performed the angular path before landing at coordinate (0.5, −1).

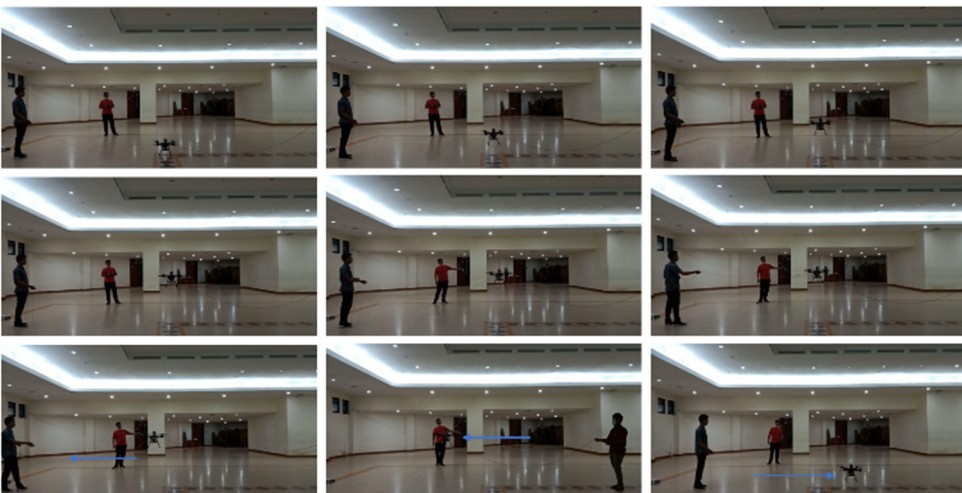

**Figure 10.** Adding disturbance by pulling the rope during hovering without activating the stability mode control.

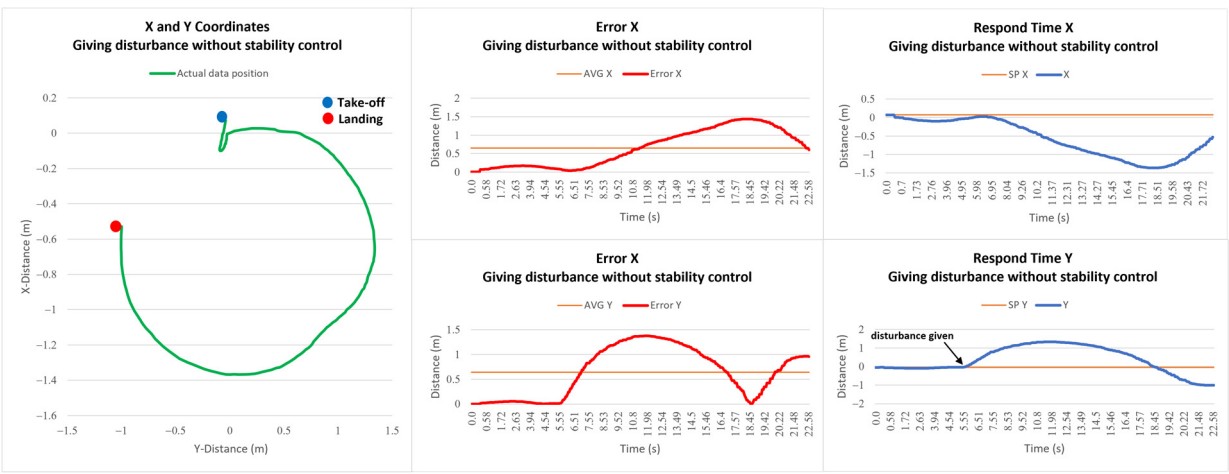

**Figure 11.** The response result from adding disturbance during hovering without stability mode control.

In comparison, we activated the stability mode control while pulling the drone during hovering, as presented in Figure 12. As we can see in Figure 12, the drone first flew at a certain height using the remote control. Then, after the remote was released in a few seconds, the drone was pulled on the Y-axis until the body tilted to the right; the drone condition after pulling the rope is depicted in the figure by the blue arrow. After a few seconds on the tilt condition, the drone succeeded in fixing its position to the original position. The graph response results are represented in Figure 13, where from the graph result, the drone was able to maintain its position after the disturbance given to it and survived the disturbance given.

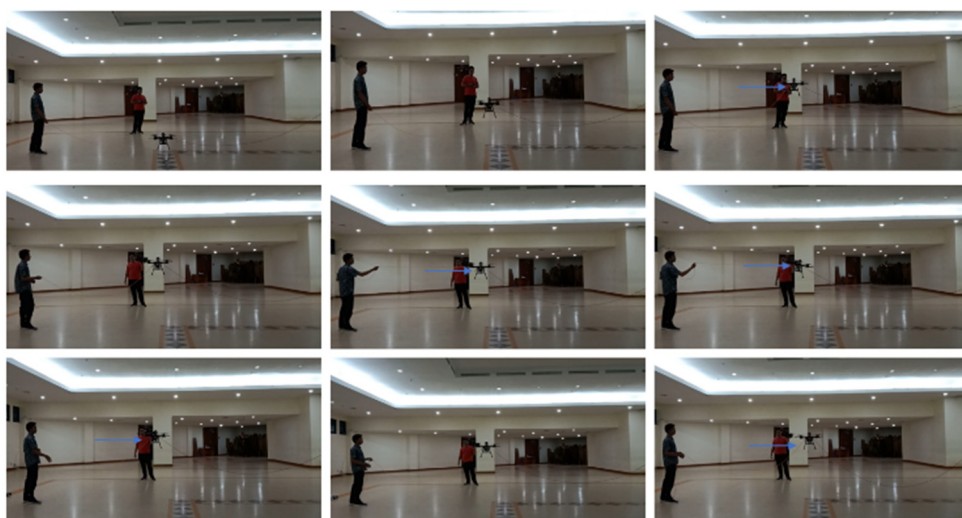

**Figure 12.** Adding disturbance by pulling the rope during hovering by activating the stability mode control.

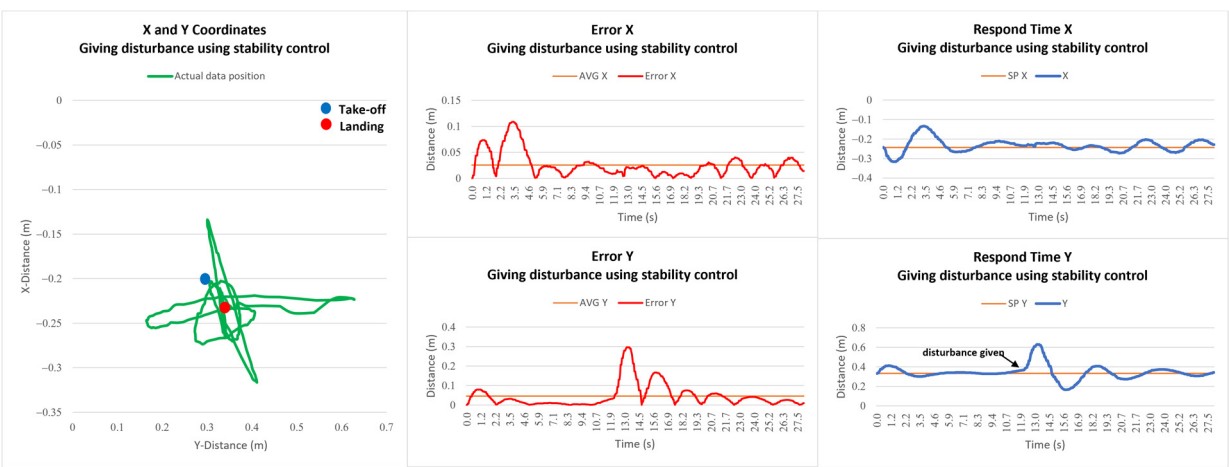

**Figure 13.** The response result from adding disturbance during hovering using stability mode control.

### 4.2. The Indoor Localization Experiments

Before conducting the indoor localization experiments, we first commanded the drone to move in certain positions and stay hovering several times by activating the stability mode control or without any activated stability mode control. Without activating the stability mode control, we commanded the drone to move from coordinate (0, 0.5) to coordinate (5, 0) and let the drone hover several times. The result of this experiment is represented in Figure 14, where we can see from the graph that the drone could move from the original point to the destination; however, it failed to stay hovering in the destination coordinate. The drone flew away from the end point coordinate and landed at the coordinate (8, −2.5); the average error generated at the X-axis is about 1.5 m and at Y-axis is approximately 1.8 m.

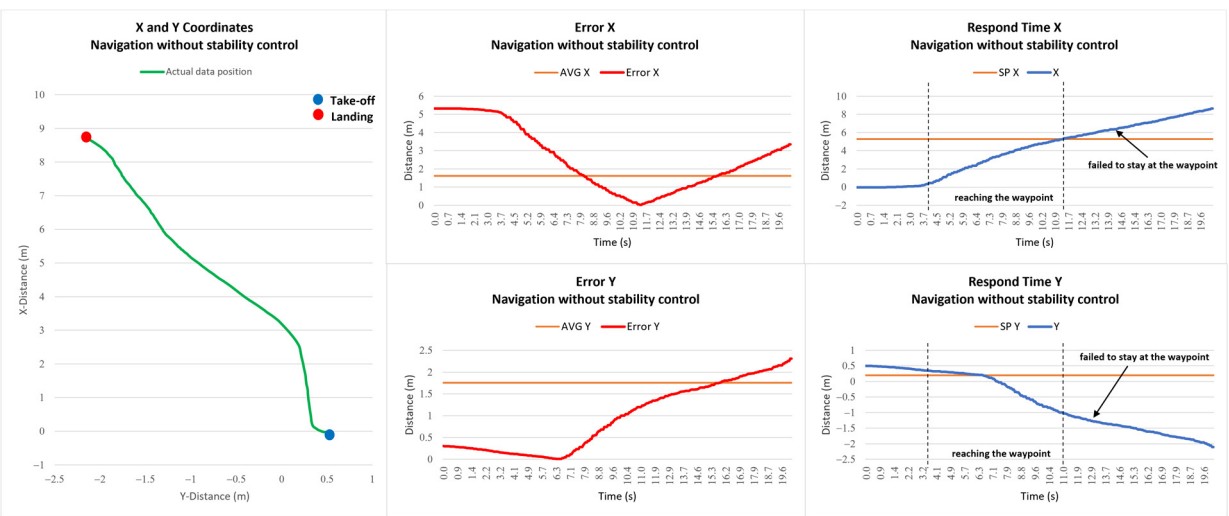

**Figure 14.** The response result of the simple indoor navigation without activating the stability mode control.

After activating the stability mode control, the drone can move from its original position to the destination as the system commands. On this occasion, we commanded the drone to move from coordinate (0, 1) to (5, 0.2). As seen in Figure 15, at first, the drone took off from its original position at coordinate (0, −1), then it tried to move to the destination at coordinate (5, 0.2) and stayed hovering in this end-point coordinate. Judging by the result given in Figure 15, it can be concluded that the drone was able to maintain its position due to the results of the average error at the X- and Y-axes at about 0 m and the drone remained hovering in this coordinate stably at a particular time. After ensuring that the drone can maintain its stability during hovering, the indoor localization practical can be verified. In this experiment, we ordered the drone to follow the rectangle and triangular path in a real-time application. To ensure that the drone moved accordingly to the path, we took the white rectangular paper with dimensions of about 40 cm × 40 cm and scattered it on the floor to resemble the triangular and rectangular path.

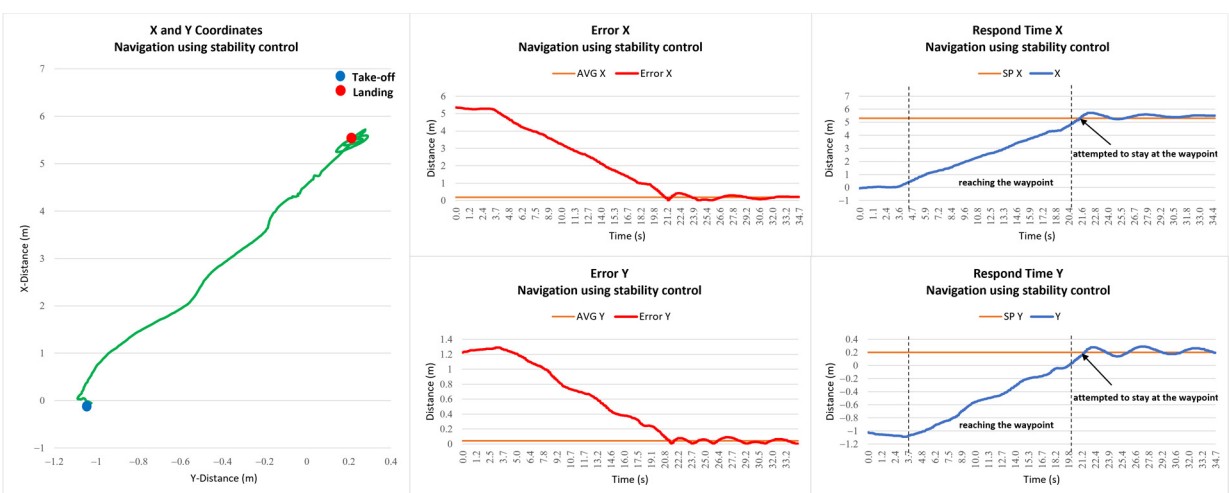

**Figure 15.** The response result of the simple indoor navigation by activating the stability mode control.

The rectangular path experiment is presented in Figure 16; in this experiment, we set the rectangular path as 5 m × 5 m using white paper indoors and let the drone be stopped and continued to hover in a particular coordinate, in this case, coordinates (0, 0), (5, 0), (5, 5), and (0, 5). As seen in Figure 16, a light spot above each white rectangular indicated that the ZED camera recognized the coordinate and commanded the drone to hover above the white

rectangular. The response of this movement is represented in Figure 17; where the blue line denotes the path generated by the drone, the orange lines show the response system, and the red one presents the error during the maneuver. As for the response towards time for the X- and Y-axis, at first, the drone will remain hovering at coordinate (0, 0), then move to coordinate (5, 0), then coordinate (0, 5), and land at coordinate (0, 0) as the original position. Besides following the rectangular path, we resembled the triangular path using the same paper, which can be seen in Figure 18. The coordinates for the triangular path are (0, 0), (5, −4), (5, 4), and (0, 0) as the original position. The response system for this navigation is represented in Figure 19, where the graph produced errors which were almost the same as those of the rectangular path.

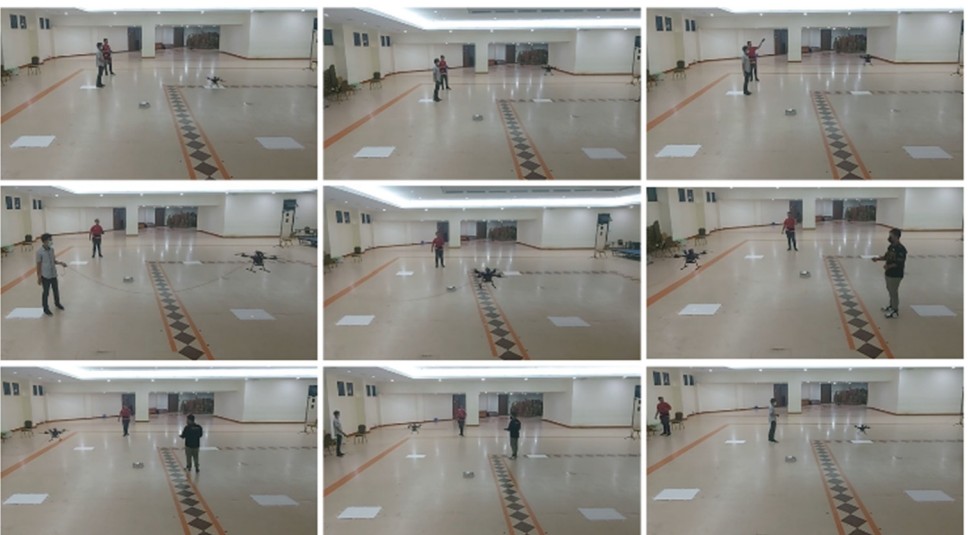

**Figure 16.** The rectangular path is to follow by the drone indoor.

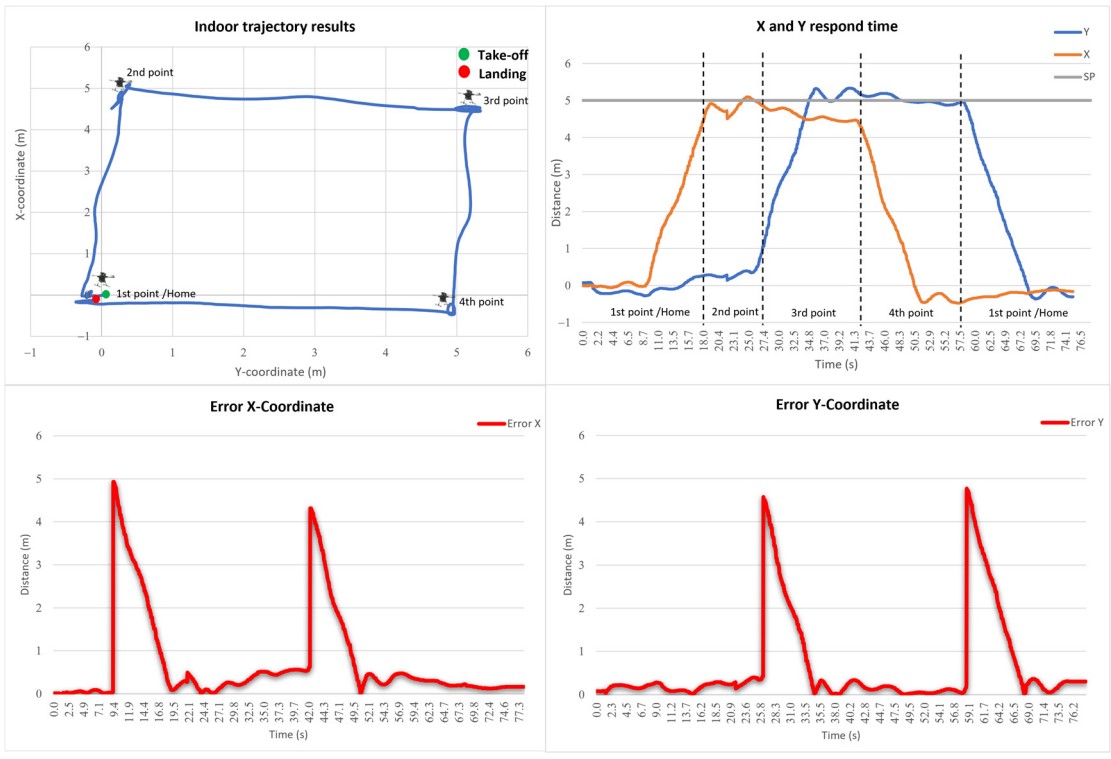

**Figure 17.** The response result of tracking the rectangular path indoors.

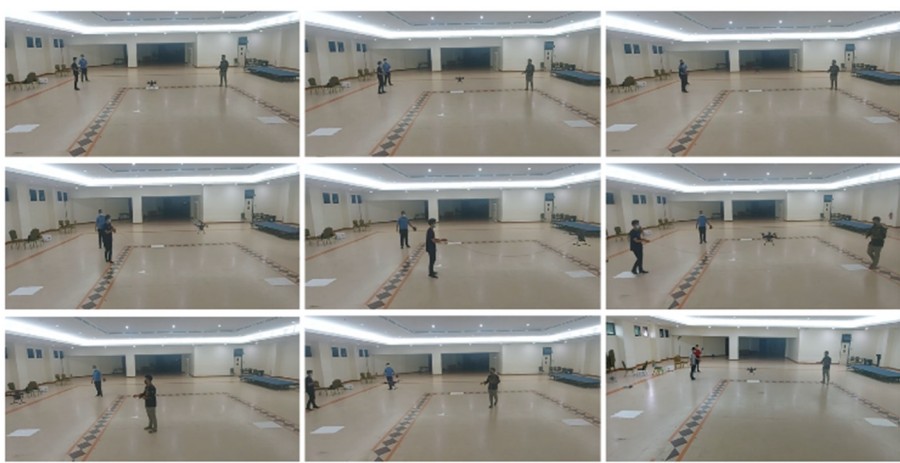

**Figure 18.** Tracking the triangular path indoor.

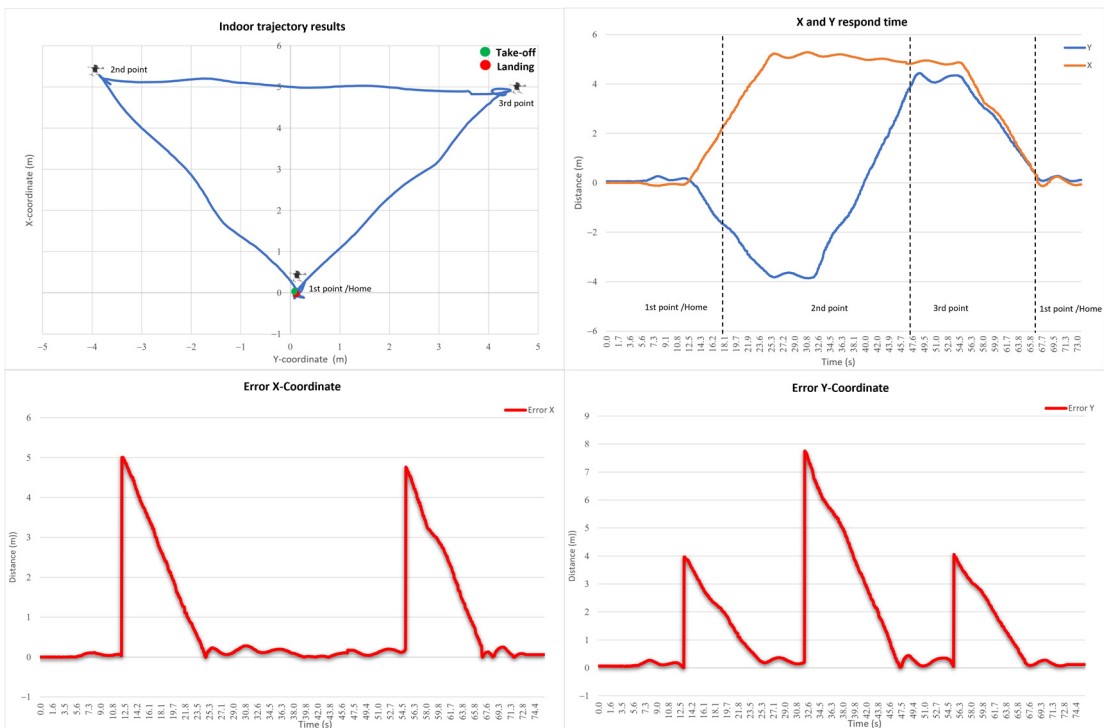

**Figure 19.** The response result of tracking the triangular path indoor.

## 5. Conclusions

This paper discussed alternative solutions to an indoor localization by using the features provided by the stereo camera. At first, we transformed the input camera coordinate into the NED system before performing the indoor localization, which aimed to control the drone's stability before being commanded to complete an indoor localization. All the experiments in this work have been done in a real-time application. The first experiment described that the drone could maintain its stability even if it added some disturbance. Moreover, for the next experiment, the drone succeeded in following both the rectangular and triangular path. However, following the path generated the error or oscillated it in particular coordinates. Therefore, in the future, we will focus on eliminating the oscillation which occurred during hovering to minimize the error.

**Author Contributions:** Conceptualization, A.R.F. and R.A.; methodology, A.R.F. and R.A.; validation, A.R.F. and R.A.; formal analysis, A.R.F. and R.A.; investigation, A.R.F. and A.H.; resources, A.H., A.S. and R.A.; data curation, R.A.; writing—original draft, R.A and A.R.F.; writing—review and editing, R.A and A.R.F.; visualization, R.A.; supervision, A.R.F. and R.A.; project administration, A.H. and A.S.; funding acquisition, A.R.F. All authors have read and agreed to the published version of the manuscript.

**Funding:** This research was fully funded by Politeknik Negeri Batam.

**Data Availability Statement:** Experimental video results can be found in #mdpi localization video.

**Conflicts of Interest:** The authors declare no conflict of interest.

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
