# Peer review of "Indoor Localization Using Positional Tracking Feature of Stereo Camera on Quadcopter"

_electronics, doi:10.3390/electronics12020406_

Round 1
Reviewer 1 Report
The Indoor Localization Using Positional Tracking Feature of Stereo Camera on Quadcopter
The paper presented a new indoor navigation approach for quadcopters using ZED 2 stereo camera and NED system.
Here are some concerns about the work:
· The manuscript needs to be improved in terms of language
· The related work section is missing and also the gaps in the existing research
· The quality of most figures need to be improved (Figure 8,9,11,13,14,15,17, and 19)
Author Response
Response to Reviewer 1 Comments
Point 1: The manuscript needs to be improved in terms of language
Response 1: We already proofread our manuscript and fixed some missing approaches.
Point 2: The related work section is missing and also the gaps in the existing research
Response 2: We already proofread our manuscript and added some references in the introduction related to the ZED 2 camera and indoor localization.
Point 3: The quality of most figures need to be improved (Figure 8,9,11,13,14,15,17, and 19)
Response 3: We already changed our picture with high resolution and replaced it with English as well for the legend and the title of the pictures.
Reviewer 2 Report
This paper describes the use of the ZED 2 (stereolabs) stereo camera to position an Quadcopter in GPS-denied environments. The main credit of this paper is coordinate transformation of the by the ZED 2 measured position, orientation and confident level of a white square paper on the floor to the pose of the drone itself. This method and approach is described in sections 2.1 (the coordinate transformation) and 2.2 (the system architecture). Results of the performed tests (experiments) are given in chapter 3.
This is an interesting approach, but the paper is not that very well written (check English grammar) and it lacks an proper insight in the functionality of the ZED2 stereo camera. in the first chapter (Introduction) hardly any (proper) references are made to other papers using this ZED2 camera. In fact, even no reference is given to https://www.stereolabs.com/ or positional tracking in general.
The abstract is very brief, and has some annoying English mistakes (This paper will present a new appoached ..), and non-explained acronyms (NED system).
The conclusions are also open for discussion, like “some components experienced an overheating condition during flight”.
Author Response
Response to Reviewer 2 Comments
Point 1: This paper describes the use of the ZED 2 (stereolabs) stereo camera to position an Quadcopter in GPS-denied environments. The main credit of this paper is coordinate transformation of the by the ZED 2 measured position, orientation and confident level of a white square paper on the floor to the pose of the drone itself. This method and approach is described in sections 2.1 (the coordinate transformation) and 2.2 (the system architecture). Results of the performed tests (experiments) are given in chapter 3.
This is an interesting approach, but the paper is not that very well written (check English grammar) and it lacks an proper insight in the functionality of the ZED2 stereo camera. in the first chapter (Introduction) hardly any (proper) references are made to other papers using this ZED2 camera. In fact, even no reference is given to https://www.stereolabs.com/ or positional tracking in general.
Response 1: We already proofread our manuscript and added some references in the introduction related to the ZED 2 camera and indoor localization.
Point 2: The abstract is very brief, and has some annoying English mistakes (This paper will present a new appoached ..), and non-explained acronyms (NED system).
Response 2: We already proofread our manuscript and changed the abstract in accordance with our topic of the journal and explain the NED system acronyms in the abstract.
Point 3: The conclusions are also open for discussion, like “some components experienced an overheating condition during flight”.
Response 3: We already revised the conclusion in accordance with the experiment result and the topic of the journal.
Round 2
Reviewer 1 Report
The manuscript has been improved but still there is a need to add/edit some sections as follows:
- The related work section needs to be added
- The figures are still very small with low resolution.
Author Response
Point 1: The related work section needs to be added
Response 1: We added the related work section in section 2.
Point 2: The figures are still very small with low resolution.
Response 2: We already upgrade the resolution of our picture in the manuscript.
Reviewer 2 Report
The paper had been proofread (should have been done before submitting the first submission). The remarks made have been improved.
Author Response
Response to Reviewer 2 Comments
we added the related work in section 2 and improved the quality of our pictures.